# Cortical recurrence supports resilience to sensory variance in the primary visual cortex

Hugo J. Ladret [1,2 ✉], Nelson Cortes[2], Lamyae Ikan[2], Frédéric Chavane [1], Christian Casanova [2] & Laurent U. Perrinet [1]

Our daily endeavors occur in a complex visual environment, whose intrinsic variability challenges the way we integrate information to make decisions. By processing myriads of parallel sensory inputs, our brain is theoretically able to compute the variance of its environment, a cue known to guide our behavior. Yet, the neurobiological and computational basis of such variance computations are still poorly understood. Here, we quantify the dynamics of sensory variance modulations of cat primary visual cortex neurons. We report two archetypal neuronal responses, one of which is resilient to changes in variance and co-encodes the sensory feature and its variance, improving the population encoding of orientation. The existence of these variance-specific responses can be accounted for by a model of intra-cortical recurrent connectivity. We thus propose that local recurrent circuits process uncertainty as a generic computation, advancing our understanding of how the brain handles naturalistic inputs.

[1] Institut de Neurosciences de la Timone, UMR 7289, CNRS and Aix-Marseille Université, Marseille, France. [2] School of Optometry, Université de Montréal, Montréal, Canada. ✉email: hugo.ladret@univ-amu.fr

Selectivity to the orientation of visual stimuli is an archetypal feature of the neurons in the mammalian primary visual cortex (V1)[1], which has been historically studied using low-complexity stimuli such as oriented gratings[2]. While this approach offers a clear hypothesis as to what neurons are responding to, it only probes for neural selectivity to individual input parameters, such as orientation or spatial frequency. Natural vision, however, involves rich cortical dynamics[3] integrating a mixture of multiple local parameters and global contextual information[4]. Hence, a majority of our understanding of V1 relies on neural responses to single inputs in orientation space, rather than naturalistic responses to multiple orientations.

This knowledge gap is not trivial, as the variance of distributions of sensory inputs is a fundamental cue on which our brain relies to produce coherent integration of sensory inputs and prior knowledge of the world[5,6] in order to drive behavior[7]. According to Bayesian inference rules, low-variance inputs are processed through fast feedforward pathways, whereas higher sensory variance elicits a slower, recurrent integration[8]. How the brain performs computations on variance is not yet fully understood. In V1, it has been shown that single neurons undergo nonlinear tuning modulations as a function of their input's variance[9] which can serve as a functional encoding scheme[10,11]. These recent results align with earlier models of recurrent cortical activity of V1[12,13] and also match psychophysical measurements in humans[14–16]. While it seems that local interactions within V1 are sufficient to encode orientation variance[17], the quantification of single neuron responses, their dynamics and their link to a functional population encoding of variance remains to be established.

Here, we investigate the neural basis of variance processes in V1 using stimuli matching the orientation content of natural images[18]. We present a quantitative analysis of single neurons' variance-tuning functions, as well as their dynamics, reporting heterogeneous modulations. Two archetypal response types emerge in V1, one of which relies on predominantly supra-granular neurons that maintain robust orientation tuning despite high sensory variance, allowing them to co-encode orientation and variance, and enhancing V1's orientation distribution encoding. A well-established V1 intracortical recurrence model accounts for these resilient neurons, aligning with canonical Bayesian frameworks[6] and suggesting uncertainty computations as a new generic function for local recurrent cortical connectivity.

## Results

**Single-neuron response in V1 depends on input variance**. We recorded neural activity from 249 anesthetized cat V1 neurons and measured orientation-selective responses to naturalistic images called Motion Clouds[18]. These stimuli are band-pass filtered white noise textures and offer three advantages over both simple grating-like stimuli and complex natural images. First, they enable fine control of mean $\theta$ and variance, controlled by $B_\theta$, of orientation distributions through a generative model, thereby reproducing natural images' oriented content (Fig. 1). Second, as they are stationary in the spatial domain, they only probe orientation space, excluding any second-order information exploitable by the visual cortex[19]. Third, by conforming to natural images' $1/f^2$ power spectrum distribution[20], they attain a desirable balance between controllability and naturalness[21]. We generated 96 Motion Clouds by varying mean orientation $\theta$ between 0° and 180° in 12 even steps and variance $B_\theta$ between ≈0° and 35° in eight evenly spaced steps.

All recorded neurons displayed orientation selectivity to Motion Clouds. Nearly all (98.8%, $p < 0.05$, Wilcoxon signed-rank test) units maintained their preferred orientation when

variance $B_\theta$ increased, while the peak amplitude of the tuning curve diminished significantly (95.1% units, $p < 0.05$, Wilcoxon signed-rank test, 73.1% mean amplitude decrease for $B_\theta = 35°$). Only 28.5% of the recorded units were still tuned for $B_\theta = 35.0°$ stimuli ($p < 0.05$, Wilcoxon signed-rank test). Thus, increasing input variance reduces single neuron tuning, which manifests heterogeneously across neurons, as evidenced by two representative single units shown in Fig. 2a. Neuron A illustrates single units which are no longer orientation-tuned when variance $B_\theta$ reaches 35° ($W = 171.0$, $p = 0.24$, Wilcoxon signed-rank test), unlike neuron B ($W = 22.5$, $p = 10^{-6}$) which exemplifies the aforementioned 28.5% variance-resilient units. These response types are characterized by functions relating $B_\theta$ to the goodness of tuning (circular variance, CV), named here variance-tuning functions (VTF, Fig. 2b). Such VTFs represent the input/output transformation in variance space, and are well-fitted with Naka-Rushton functions[22] (Supplementary Fig. 2a). This allows to summarize variance modulations using only three parameters: $n$, the VTF non-linearity; $B_{\theta 50}$, the input variance level for the tuned-untuned state transition; and $f_0$, the orientation tuning goodness for lowest-variance inputs. Overall, VTFs exposed diverse responses to variance among V1 neurons, with median values outlining a characteristic VTF that is slightly nonlinear, with a changepoint at $B_\theta = 19.2°$ (Fig. 2c). In other words, most neurons tend to change abruptly in tuning when input variance reaches 19.2°, after which the response becomes less sensitive to orientation. Alternative metrics were also calculated, including variance-half width at half height (HWHH) and variance-maximum response functions (Supplementary Fig. 2b–e). Although HWHH displayed patterns resembling VTFs, we elected to not use it, as its reliance on fits, its consequent susceptibility to fitting artifacts, and its similarity with CV are not desirable properties. Since CV also inherently accounts for the firing rate at the preferred orientation (see "Methods"), we relied on this metric to describe both maximum amplitude and goodness of tuning in a single metric.

Orientation variance impacts not only orientation tuning but also the dynamics of the response of V1 neurons (Fig. 3). Interestingly, both effects are linked, as demonstrated by the two example VTFs: neuron B, which exhibited orientation-tuned responses for $B_\theta = 35°$ inputs (Fig. 2a), also had a slower time-dependent change of goodness of tuning (relative min. of reduction of 42% of max. CV at 200 ms post-stimulation onset, $B_\theta = 0°$) compared to neuron A (relative min. of 26% of max. CV at 90 ms post-stimulation onset, Fig. 3b). These dynamical modulations were also heterogeneously distributed among the population, significantly more spikes emitted 200 ms after stimulation onset for $B_\theta = 35°$ (Fig. 3d, $U = 14936.0$, $p < 0.001$, Mann–Whitney $U$-test). In summary, orientation variance induces changes in both tuning and dynamics of V1 neurons, revealing two archetypal types of response: either fast in time and nonlinear with respect to variance (neuron A) or slow in time and linear with respect to variance (neuron B).

**Multiple types of variance responses are found in V1**. To properly characterize the two aforementioned types of responses to variance, we separated the recorded neurons into two groups using K-means clustering the Principal Components (PC, Fig. 4) of the neuronal responses. Clustering was performed on the VTFs (Fig. 4b), tuning statistical measurements (Fig. 4c, d) and response dynamics (Fig. 4e, f). We used the first 2 PC for clustering the data, which accounted for 39.1% of the cumulative variance (Supplementary Fig. 4a), and chose two clusters based on the number of example responses and the empirical absence of an elbow[23] in the Within-Clusters-Sum-of-Squares (WCSS) curve (Supplementary Fig. 4b). This splits the data into a cluster of 164

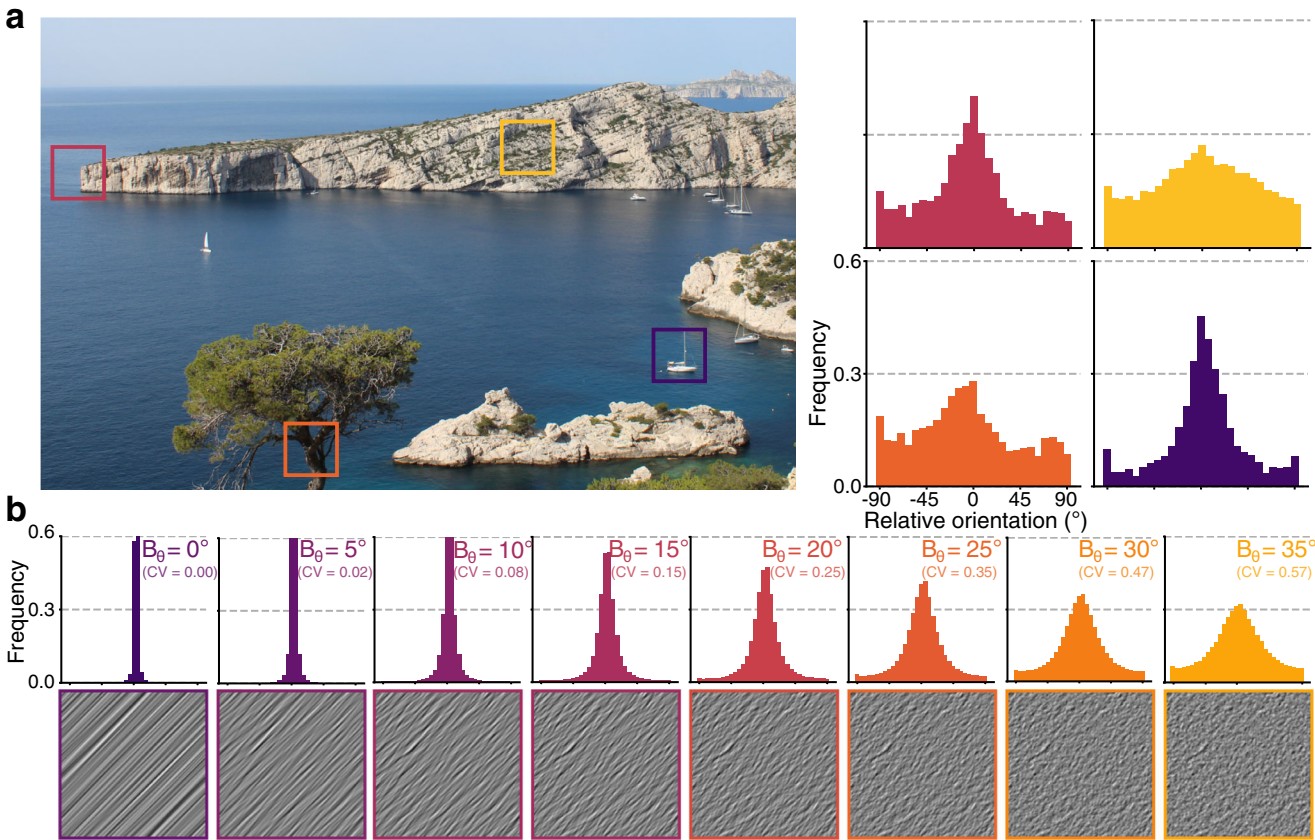

**Fig. 1 Variance of orientation distributions characterizes local regions of natural images. a** Distributions of orientation from four 200x200 px regions of a natural image (picture taken by H.J.L.) obtained by a histogram of oriented gradients (32x32 px/cell), centered around the most frequent orientation. **b** Motion Clouds, naturalistic stimuli (bottom row) with mean orientation $\theta = 45°$ and increasing variance ($B_\theta$) from left to right. Distributions of the orientation of the stimuli are shown on the upper row. Circular Variance (CV) of the distribution is shown for comparison.

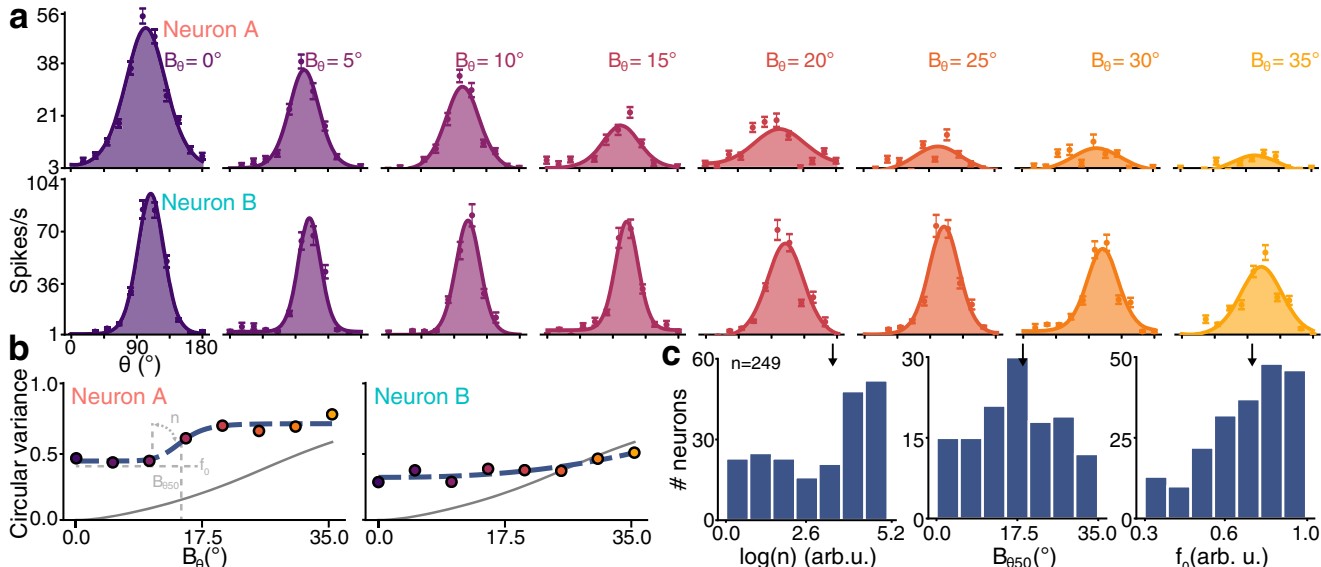

**Fig. 2 Single-neuron tuning correlates with input variance.** Additional examples are shown in Supplementary Fig. 1. **a** Tuning curves of two neurons responding to Motion Clouds of increasing variance $B_\theta$. Dots indicate the mean firing rate across trials (baseline subtracted, 300 ms average), error bars are the standard error and lines represent a fitted von Mises function. **b** Variance-tuning functions (VTF), relating the change of orientation tuning measured by the circular variance (CV, dots) as a function of input variance $B_\theta$, fitted with a Naka-Rushton (NKR) function (dashed curves, parameters shown in light gray). Parameters of the VTF are $\log(n) = 8.4$, $B_{\theta 50} = 14.7°$, $f_0 = 0.4$ for neuron A and $\log(n) = 2.4$, $B_{\theta 50} = 35.0°$, $f_0 = 0.3$ for neuron B. The CV identity curve is shown in solid gray. **c** Histograms of the NKR parameters (in the [5%;95%] range of possible NKR fitting values) for the 249 recorded units. Median values are indicated by a black arrow ($\log(n) = 3.6$, $B_{\theta 50} = 19.2°$, $f_0 = 0.75$).

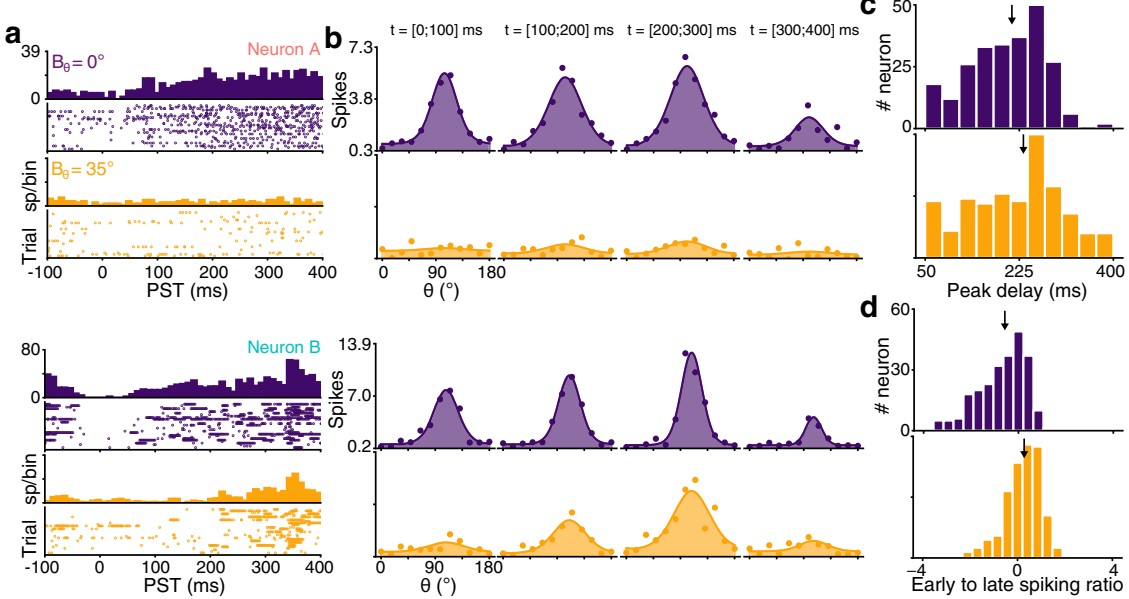

**Fig. 3 Neural dynamics depend on input variance.** Additional examples are shown in Supplementary Fig. 3. **a** Peristimulus time (PST) histogram and rasterplot for the two previous example neurons, at variance $B_\theta = 0°$ (purple) and $B_\theta = 35°$ (yellow). **b** Tuning curve dynamics in 100 ms windows, starting at labeled times. **c** Delay to peak amplitude of tuning curves for $B_\theta = 0°$ (purple, median = 210 ms) and $B_\theta = 35°$ (yellow, median = 233 ms) for the population. Median values are indicated by a black arrow. **d** Log ratio of early (<100 ms post-stimulation) and late (>200 ms) spike counts for $B_\theta = 0°$ (median = −0.54) and $B_\theta = 35°$ (median = 0.27) for the population.

neurons, including neuron A, and another cluster of 85 neurons associated with neuron B's response type. As neuron B displayed resilience to increased input variance (Fig. 2a), its cluster was labeled resilient neurons. Conversely, neurons clustered with neuron A were labeled vulnerable neurons (blue and red colors, respectively, Fig. 4a). Opting to categorize the data into two distinct response types facilitates a comprehensive understanding of the underlying continuum of behaviors. This approach has proven successful in the characterization of novel visual responses, such as V1 simple/complex cells[24] and MT pattern/component cells[25].

The K-means clustering resulted in a significant difference between the two groups' VTF parameters (Fig. 4b): resilient neurons had significantly more linear modulations ($\log(n)$, $U = 4029.0.0$, $p < 0.001$, Mann–Whitney $U$-test), higher changepoints ($B_{\theta 50}$, $U = 7854.0$, $p = 0.028$) and better tuning to low-variance inputs ($f_0$, $U = 4992.0$, $p < 0.001$), which endows them with the ability to respond to an orientation on a broader range of input variances[26,27]. No significant differences in the variance-HWHH and variance-firing rate functions were observed, except for the non-linearity of the latter metric (Supplementary Fig. 5). This is coherent with the clustering on the statistical measurement of orientation tuning, which showed that resilient neurons remained significantly tuned to higher values of $B_\theta$ ($B_{\theta max}$, Fig. 4c, $U = 9155.0$, $p < 0.001$). However, both groups of neurons had a similar circular variance for $B_\theta = 35°$ (Fig. 4d). This suggests that both types of neurons were similarly poorly tuned for inputs of the highest variance, but underwent different tuning changes between $B_\theta = 0°$ and $B_\theta = 35°$. In terms of dynamics, the two groups exhibited the same differences that characterized neurons A and B. Resilient neurons discharged significantly later than vulnerable neurons for $B_\theta = 0°$ (Fig. 4e, $U = 8455.5$, $p = 0.002$), but both groups were on par for inputs of $B_\theta = 35°$ ($U = 7794.5$, $p = 0.063$). Interestingly, resilient neurons had significantly lower time to the maximum amplitude of the tuning curve for $B\theta = 0°$ (Fig. 4f, $U = 5542.5$, $p = 0.014$), which opposes the early/late ratio of spikes. Neither group showed

variance-dependent modulation of the delay to maximum spike count ($U = 3058.0$, $p = 0.084$ and $U = 11545.5$, $p = 0.090$ for resilient and vulnerable, respectively), and both groups showed similar delay for $B_\theta = 35°$ ($U = 6094.5$, $p = 0.158$).

The existence of these two groups of neurons could not be attributed to the integration of the drifting motion of the stimuli (direction selectivity index, unused in the clustering process, Fig. 4g, $U = 7031.5$, $p = 0.910$). Instead, the location of the recorded units (unused in the clustering process) predominantly positioned the resilient neurons in supragranular layers, offering a mechanistic basis for their existence (Fig. 4h). Moreover, resilient neurons have sharper orientation tuning and slower dynamics, which are distinctive features of supragranular neurons[28,29]. This, however, does not establish a functional role for these two types of responses in V1.

**Population-level modulations of the orientation code.** As the neuronal population has been separated into well-characterized groups, we wish to understand the functional role played by resilient and vulnerable neurons. To that end, we used a neuronal decoder that probes for population codes in V1, enabling us to seek what parameters of the stimuli each neuron group was encoding. We trained a multinomial logistic regression classifier[30], a probabilistic model that classifies data belonging to multiple classes (see "Methods"). This classifier received the firing rate of neurons in a sliding time window (100 ms) and learned, for each neuron, a coefficient that best predicts the class (i.e., the generative parameter $\theta$, $B_\theta$ or $\theta \times B_\theta$) of the stimulus.

This decoder was first used to probe for representation of the stimuli's orientations $\theta$ in the population activity. For this purpose, the dataset of trials was separated for each variance, such that eight independent, $B_\theta$-specific, orientation decoders were learned, with optimal parametrization (Supplementary Fig. 6). These orientation decoders were able to retrieve the correct stimulus' $\theta$ well above the chance level (1 out of 12 orientations, max. accuracy = 10.56 and 4.68 times chance level for $B_\theta = 0°$ and $B_\theta = 35°$, respectively) from the entire population recordings.

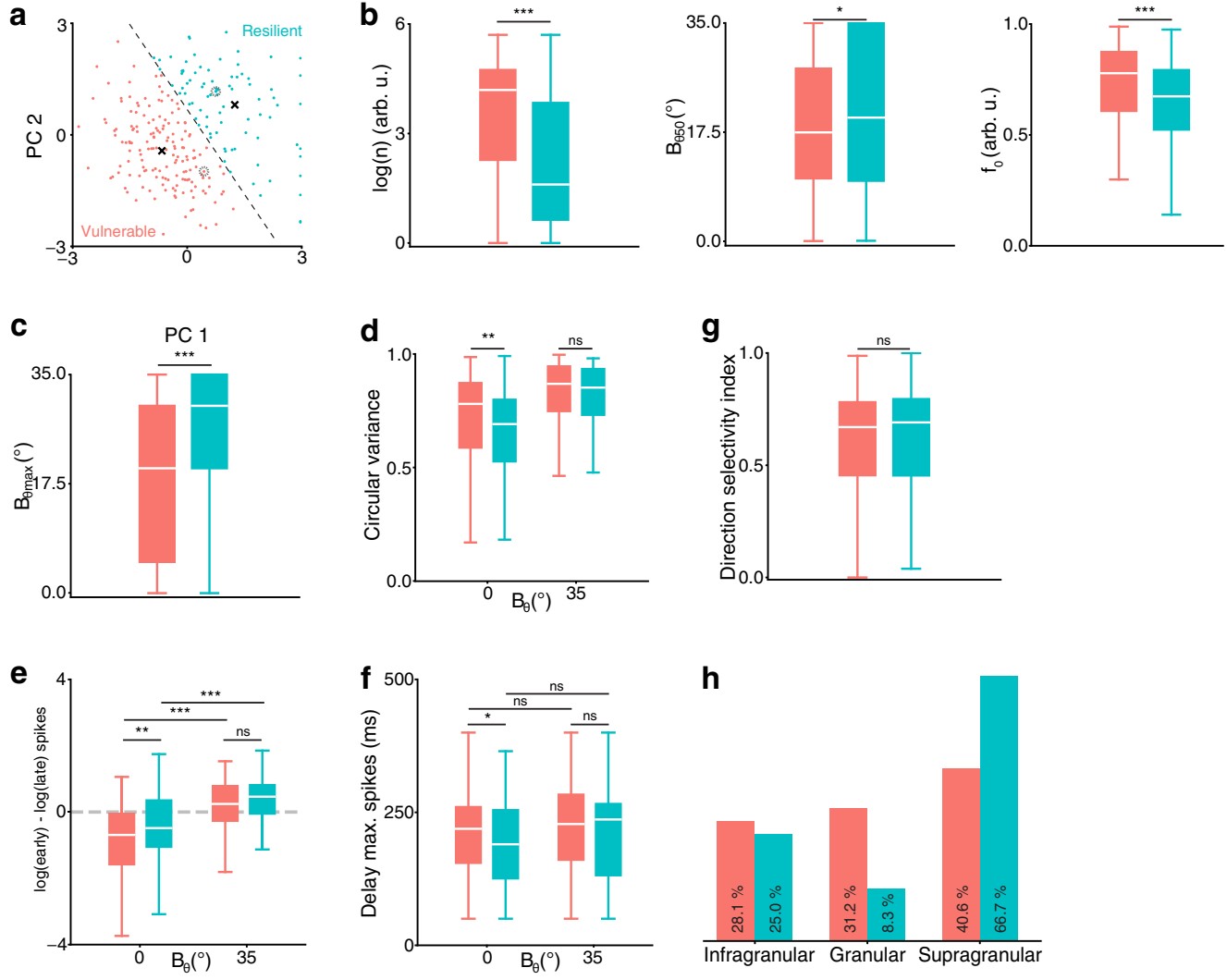

**Fig. 4 Responses to changes in variance fall into two categories. a** Principal Components (PC) analysis of the data, K-Means clustered (2 clusters, centroids shown as black crosses and separatrix as dashed line). Nine resilient neurons with PC1 > 3 are plotted at PC1 = 3. Neuron A and B are shown as dashed circles. **b** Boxplot of the VTF parameters $\log(n)$, $B_{\theta 50}$, $f_0$ (ns, not significant; *, $p < 0.05$; **, $p < 0.01$; ***, $p < 0.001$ Mann–Whitney $U$-test). Boxes cover quartile values with a median white line. Whiskers extend to $Q1 - 1.5*IQR$ and $Q3 + 1.5*IQR$, where $Q1;Q3$ are lower, upper quartiles and $IQR$ is the interquartile range. **c** Maximum $B_\theta$ for significant orientation tuning curve. **d** Circular variance at $B_\theta = 0°$ and $B_\theta = 35°$. **e** Log ratio of the early (<100 ms) and late (>200 ms) spike counts at $B_\theta = 0°$ and $B_\theta = 35°$. **f** Delay to maximum peak amplitude of tuning curves at $B_\theta = 0°$ and $B_\theta = 35°$. **g** Direction selectivity index (unused in the clustering). **h** Laminar position (unused in the clustering).

The temporal evolution of these decoders' accuracy (Fig. 5a) showed that maximally accurate orientation encoding correlates almost linearly with the stimuli's variance, as does the time to reach this accuracy (Fig. 5e, black). These dynamics depend on the input's variance, exhibiting a rapid initial rise followed by a plateau for low-variance inputs, while steadily increasing linearly over time for high-variance inputs. Interestingly, the decoding accuracy remained stable for approximately 100 ms even after a stimulus was no longer displayed. Since the decoders are trained independently in each time window, this accumulative process occurs in the recordings themselves, and not in the decoder.

The full output of these decoders (see "Methods") is a population tuning curve, which displays the likelihood of decoding all possible input classes (here, all $\theta$, Fig. 5b), rather than the proportion of correct decoding reported by the accuracy metric. The clear correlation between the sharpness of these population tuning curves (Fig. 5f left) and the accuracy of the decoder show that improvements in decoding accuracy rely directly on a population-level separation of features within orientation space[30], particularly at higher $B_\theta$ (Fig. 5b, third panel). Overall, $B_\theta$ influences the temporality of the orientation code in V1, which echoes its influence on single-neuron dynamics (Fig. 3). The short delay required to process precise inputs is congruent with the feedforward processing latency of V1[31], while the increased time required to reach maximum accuracy for low precision oriented inputs suggests the involvement of a slower, recurrent mechanism.

We then sought to assert the role of the vulnerable and the resilient neural populations by decoding $\theta$ from either group. The number of neurons in each group was imbalanced (79 more vulnerable neurons), which influences the accuracy of the decoder (Supplementary Fig. 6). Consequently, we randomly selected (with replacement) groups of 100 neurons from either population, repeating the selection 5 times. Using the same approach as with the global population decoding, we then trained $B_\theta$-specific orientation decoders on the activity of either group of neurons. Resilient neurons outperformed vulnerable ones in decoding accuracy for 56% of the time steps, mainly in the 160–330 ms

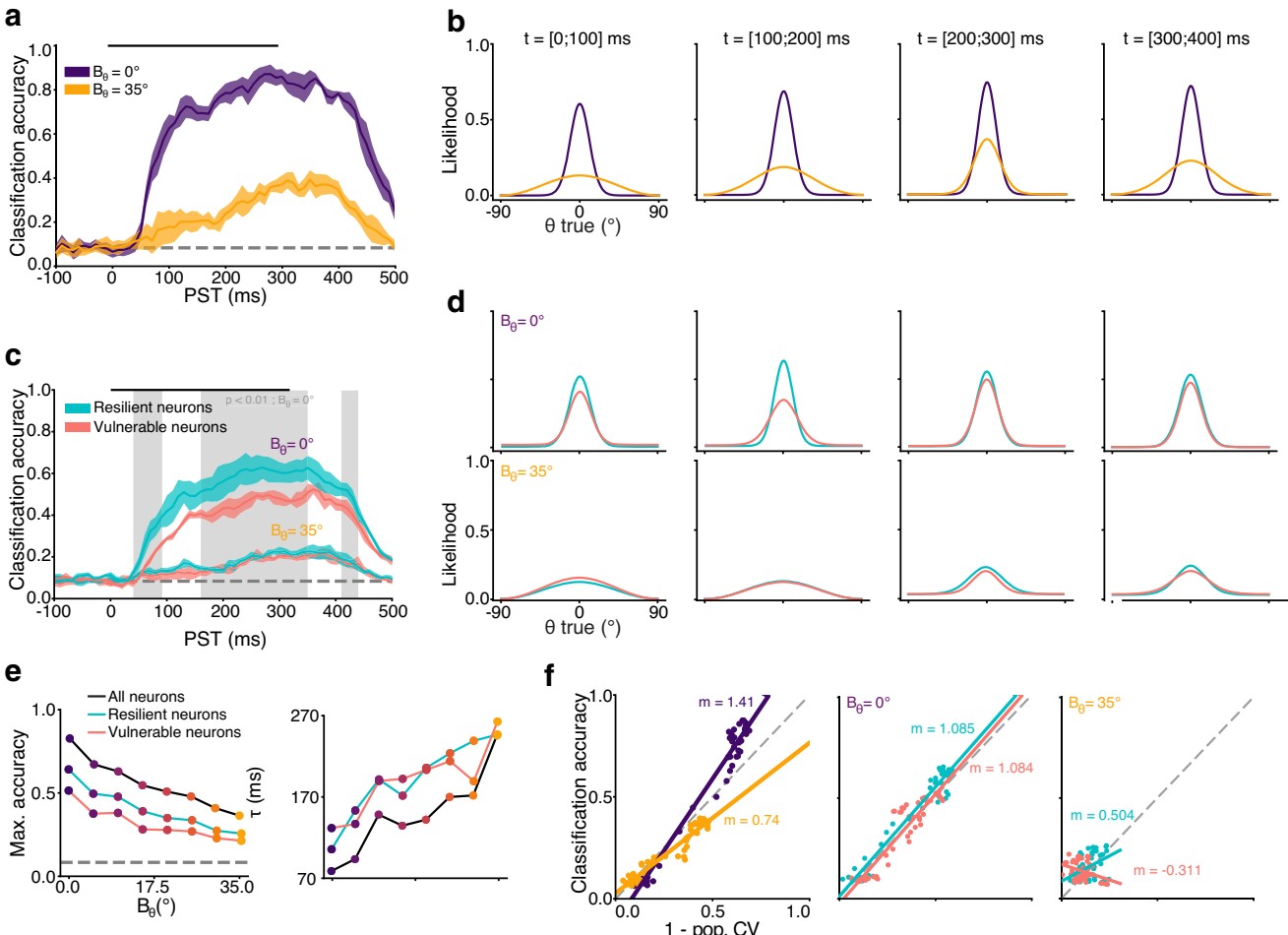

**Fig. 5 Input variance modulates orientation decoding in V1. a** Time course of orientation $\theta$ decoding accuracy at two variances $B_\theta$. Lines are the mean accuracy of $n = 5$ random resampling of 100 neurons and contour the SD. Significantly better decoding from resilient neurons at $B_\theta = 35°$ is shown as a gray overlay (Wilcoxon signed-rank test, threshold $p < 0.01$). Decoding at chance level is represented by a gray dashed line and stimulation time by a black line. **b** Population tuning curves with a von Mises fit, showing the likelihood of decoding each $\theta$ in four time windows. **c** Same as (**a**) for the two groups of neurons. **d** Same as (**b**) for the two groups of neurons, with $B_\theta = 0°$ (upper row) and $B_\theta = 35°$ (lower row). **e** Time course parameters for three decoders at all $B_\theta$, estimated by fitting a sigmoid up to PST = 300 ms. $\tau$ is the time constant. **f** Correlation between classification accuracy and population circular variance for the whole population (left), for both groups with $B_\theta = 0°$ (middle) and $B_\theta = 35°$ (right). Linear regression is shown as solid lines with slope $m$ indicated (all significant, $p < 0.001$, Wald Test with t-distribution).

period (Fig. 5c). However, both groups exhibited similar population tuning curves (Fig. 5d) and time courses (Fig. 5e). Despite the better tuning of resilient neurons to inputs with higher variance (Fig. 4), both groups have overall similar orientation encoding performances for $B_\theta = 35°$. Therefore, orientation can be decoded somewhat more effectively from the resilient neurons at the population level, but neither group appears to have a clear or stable advantage over the other in this regard, especially at higher $B_\theta$.

**A subset of V1 neurons co-encode orientation and its variance.** Given that orientation encoding did not reveal a fundamental difference in the respective contributions of resilient and vulnerable neurons, we then investigated the encoding of the stimulus' variance $B_\theta$. The same type of decoder previously used failed to infer the variance $B_\theta$ (chance level = 1 out of 8 values of $B_\theta$, max. accuracy = 1.91 times chance level) from the population activity (Supplementary Fig. 8a, b). This variance decoding also failed to reach more than twice the chance level (max. accuracy = 1.72 and 1.71 times chance level for resilient and vulnerable

neurons, respectively) in both resilient and vulnerable neurons (Supplementary Fig. 8c,d). At the single neuron level, tuning curves flatten with increments of variance (Supplementary Fig. 2a), which makes it difficult to distinguish activity generated by stimuli with $B_\theta = 0.0°$ and orthogonal orientation from the activity generated by stimuli with $B_\theta = 35.0°$ and preferred orientation. This limitation could potentially stem from the recording scale (249 neurons), which is more than an order of magnitude smaller than the quantity of neurons a single V1 biological decoder can access[32]. Thus, neither the decoding of variance $B_\theta$ nor the decoding of orientation $\theta$ accounts for a different role between resilient and vulnerable neurons.

The decoding methods used so far have assumed that V1 encodes independently single input parameters. However, a more realistic assumption is to consider the visual system's natural inputs as distributions of information (Fig. 1) that cortical neurons must process from thalamic inputs[33] based on a probabilistic computational principle[34]. Here, this implies that the naturalistic form of processing for a V1 neuron would be co-encoding both the mean feature ($\theta$) and its associated variance ($B_\theta$) to access the entire probability distribution.

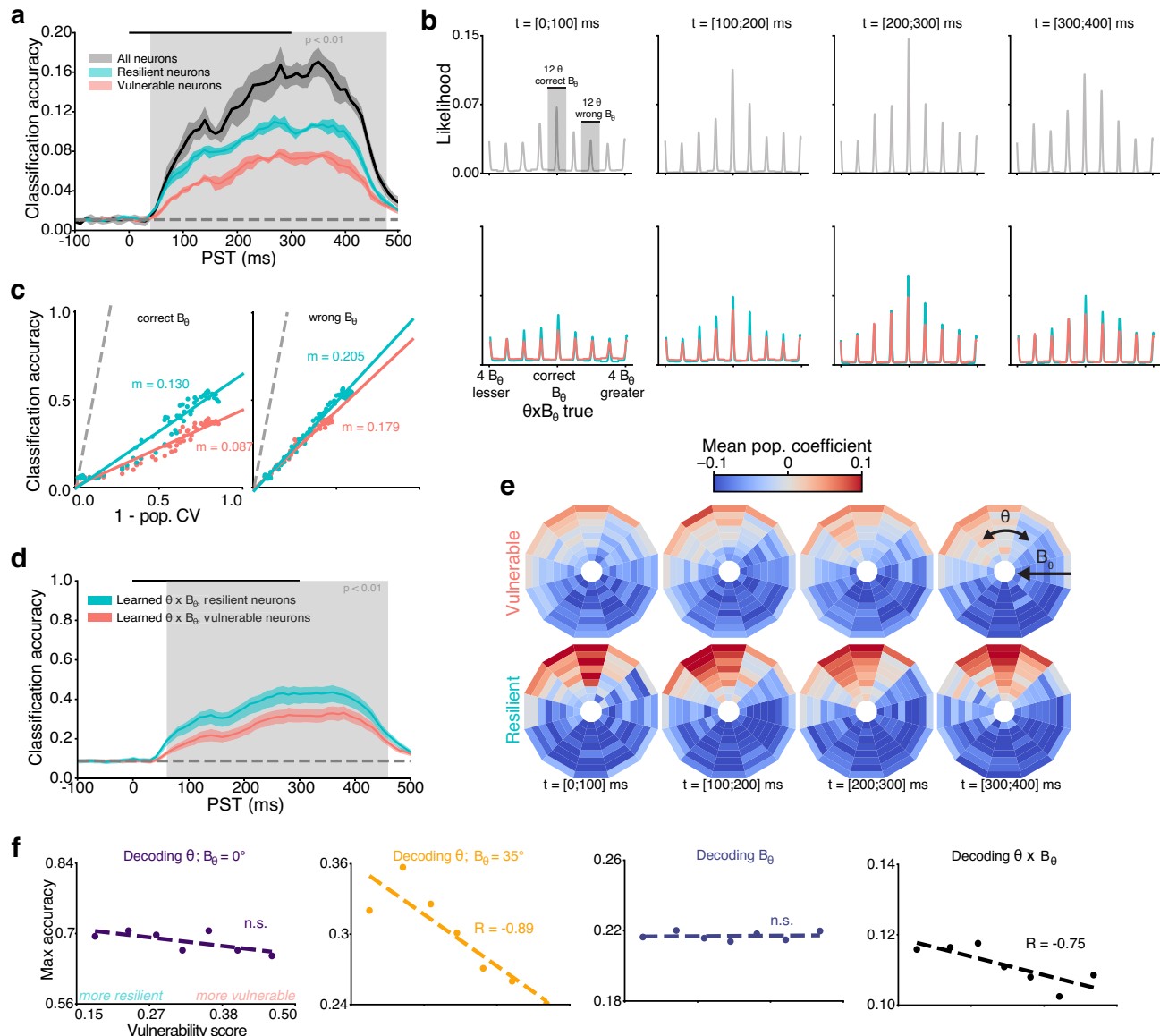

**Fig. 6 Orientation and its variance can be decoded from resilient neurons. a** Time course of the accuracy for decoding $\theta \times B_\theta$ of Motion Clouds. Lines are the mean accuracy and contour the SD. Significantly better decoding from resilient neurons is shown as a gray overlay (Wilcoxon signed-rank test, threshold $p < 0.01$). Decoding at chance level (1/96) is represented by the gray dashed line. **b** Population tuning curves for the likelihood of decoding each $\theta \times B_\theta$ in four time windows, centered around the correct $\theta \times B_\theta$. **c** Correlation between classification accuracy and population circular variance for correct $B_\theta$ population tuning curves (left) and averaged across other $B_\theta$ tuning curves (right). Linear regression is shown as solid lines with slope $m$ indicated (all significant $p < 0.001$, Wald Test with t-distribution). **d** Time course of the $\theta \times B_\theta$ decoder, marginalized over $B_\theta$ to produce $\theta$-only outputs. **e** Mean decoding coefficients of the two groups yielded from the whole population $\theta \times B_\theta$ decoder. **f** Score-based decoding for $\theta$ (first and second columns), $B_\theta$ (third) and $\theta \times B_\theta$ (fourth). Raw scores (points) are fitted with a linear regression (dashed curve), with Spearman R shown in the case of a significant correlation ($p < 0.05$).

We thus proceeded to train a decoder that retrieves both orientation and variance of the stimulus' simultaneously, referred to as a $\theta \times B_\theta$ decoder. This decoder correctly predicted orientation and variance with a maximum accuracy reaching 16.36 times the chance level (1/96, Fig. 6a, gray). The likelihood structure (Fig. 6b, upper row) showed that the correct $\theta$ was decoded alongside multiple concurrent hypothesis over $B_\theta$. The progressive increase of accuracy stems from the emergence of a dominant encoding of $\theta$ at the correct $B_\theta$, consequently diminishing the relative magnitude of representations over other $B_\theta$ values over time. Interestingly, resilient neurons showed here a different functional role from vulnerable neurons, with markedly better co-encoding of $B_\theta$ and $\theta$ (max. accuracy = 11.0 and 9.0

times chance level for resilient and vulnerable neurons, respectively, Fig. 6a, blue, red). Both groups displayed ambiguity regarding $B_\theta$ (Fig. 6b, lower row), and correlated sharpening/accuracy ratios on the correct $B_\theta$ population curve (Fig. 6c, left) or on the off-median population curves (Fig. 6c, right).

To understand the utility of this co-encoding, we marginalized the decoder over $B_\theta$, creating an orientation-only encoder that simultaneously learned both orientation and variance. Data from resilient neurons then provided significantly better encoding of orientation than vulnerable neurons (max. accuracy = 6.0 and 5.4 times the 1/12 chance level for resilient and vulnerable neurons respectively, Fig. 6d, gray regions), demonstrating that the overall V1 orientation code improves with a co-decoding of its variance.

The distinction between resilient and vulnerable neurons is further emphasized by the decoder coefficients, which represent the contributions of each type of neuron toward the overall $\theta \times B_\theta$ code (Fig. 6e, for single neuron examples see Supplementary Fig. 9). Here, these coefficients are depicted as a polar plot, where the orientation $\theta$ (centered around preferred orientation) is shown as the angle of each bin from the upper vertical and the variance $B_\theta$ is represented as the eccentricity of each bin from the center. Visualizing the coefficients of the whole population decoder (i.e., trained on the 249 neurons, Fig. 6a, gray) shows that the output learned from resilient neurons concurrently informs about both a wide range of orientations and variances, as observed by the extent of the bins in the eccentricity ($B_\theta$) axis (Fig. 6e, bottom row). On the other hand, the decoding process extracted orientation information on a very small range of $B_\theta$ from the activity of vulnerable neurons (Fig. 6e, top row). Even though the coefficients are learned independently at each time step, the difference in information between the two groups of neurons remains extremely stable through time.

Overall, orientation and its variance can be co-decoded simultaneously from resilient neurons, while only orientation can be decoded from vulnerable neurons. This is confirmed by a continuous score-based decoding metric based on the K-means parameters (Fig. 6f) that correlates, for the entire population (i.e., without splitting into two groups), their maximum decoding accuracy to a degree of vulnerability/resilience. After providing this functional rationale for resilient and vulnerable neurons, we finally address the question of how both types of neurons can exist in V1.

**Recurrent activity can explain the existence of neurons co-encoding orientation and variance**. A notable difference between vulnerable and resilient neurons is their different location within the cortical layers (Fig. 4h). This typically implies differences in local circuitry, particularly in the intra-V1 recurrent interactions between cortical columns, which are mostly confined to supragranular layers[35]. Given that resilient neurons are predominantly found in these supragranular layers, we aimed to find a mechanistic rationale for the existence of the two groups of neurons based on local interactions in V1. We developed a neural network from a well-established computational model of recurrent connectivity in V1, originally used to account for the intracortical activity in cat V1[36] and later simplified as a center-surround filter in the orientation domain[29]. This model has already accounted for an extensive range of emerging properties in cortical circuits[37,38]. Briefly, it is built of orientation-selective neurons tiling the orientation space and connected among themselves via recurrent synapses which follow an excitatory/inhibitory difference of von Mises distributions (Fig. 7a). Here, we model inputs with higher variance as more spread in orientation space (Fig. 1) and thus in model space, which hence drives the recurrent dynamics of the model based on $B_\theta$ (for a full description, see "Methods").

Considering that feedforward connectivity with heterogeneous tuning can encode mixtures of orientations and natural images[9], we first ran our model without recurrent synapses. We reproduced the heterogeneous selectivity by convolving the input with tuning curves of varying bandwidths (Fig. 7b, inset). This feedforward mode of the network was only able to produce a limited number of responses (Fig. 7b), in which increasing the bandwidth of the tuning curves increased the parameter $f_0$ of the VTF, but kept $n$ and $B_{\theta 50}$ constant.

Barring that explanation, we focused on the role of recurrent synapses and disabled the convolution of inputs. We varied the concentration parameters of the synaptic distributions $\kappa_{inh}$ and $\kappa_{exc}$ (Fig. 7c, e) in 200 even steps ranging from 0.35 to 7, yielding 40,000 possible configurations of the model. This allowed to manipulate the VTF and to accurately reproduce those of single neurons recorded in V1 (neuron A, B in Fig. 2b and C in Supplementary Fig. 1, modeled in Fig. 7c). Altering the type of recurrence between neurons with different orientation preference allowed to reproduce all VTF found in V1. The parameter spaces (Fig. 7e) showed a trend for resilient VTFs (low $n$, high $B_{\theta 50}$, low $f_0$) to be found mostly around the $K_{exc}$; $K_{inh}$ identity line, thus produced by balanced recurrent connectivity. Vulnerable VTFs (high $n$, low $B_{\theta 50}$, high $f_0$) were, on the contrary, mostly found above the identity line, where the configuration of the network is dominated by excitation over inhibition. This is consistent with the range of parameters that yielded higher response latency (Fig. 7d), which also occupied more parameter space when input variance increased. In summary, recurrence between V1 neurons seems to be sufficient to explain the existence of vulnerable and resilient neurons and, consequently, to account for the co-encoding of orientation and variance.

## Discussion

The variance of oriented inputs to V1 impacts orientation selectivity[9] and we have sought to understand how V1 could process this input parameter. We found that variance causes modulations in tuning (Fig. 2) and dynamics (Fig. 3) of single V1 neurons, which we have classified as either vulnerable or resilient (Fig. 4). Decoding analysis revealed variance-dependent accumulative dynamics in the two groups of neurons (Fig. 5) that are directly tied to a population-level separation of features within orientation space[30]. Both groups can encode orientation but not variance (Supplementary Fig. 8), and only resilient neurons are able to accurately co-encode orientation and variance of the input to V1 (Fig. 6). Based on cortical layer position (Fig. 4h) and on a computational approach (Fig. 7), we propose that the processing input variance in V1 is supported by recurrent connectivity between local cortical populations (Fig. 8). This not only improves the encoding of orientation in V1 but also links directly to canonical Bayesian frameworks, suggesting uncertainty computation as a new mechanism supported by local recurrent cortical connectivity.

Here, we restricted our approach to orientation space, rather than investigating the full extent of spatial relationships which are present in natural images. Thus, full-field stimuli without second-order correlation were used, which compared to a purely ecological environment, have likely excluded end-stopped cells[39]. While this approach limited the responses to V1 and excluded higher-order cortical areas, there exists both neuro-biological and computational evidence that V1 does not need to recruit other cortical areas to process orientation variance. For instance, the heterogeneous recurrent excitatory and inhibitory synaptic connectivity in V1[40–43] sustains resilient orientation tuning[44] that can account for the diversity of single neurons' resilience under different connectivity profiles, as explored in our computational model (Fig. 7). This is supported by the temporal scale of local recurrent connectivity, namely the slowly-conducted horizontal waves in an orientation map[45], which fit the view of variance processing as an iterative and accumulative computation implemented by local recurrent interactions between supragranular resilient neurons that are heavily connected through recurrent interactions with neighboring cortical columns[28,29,35,45]. In this regard, our reported time scales may have been slightly affected by the use of anesthesia (halothane), which has a limited visible effect on V1[46,47] and is less likely to cause modulations in this area compared to higher-order areas[48–51].

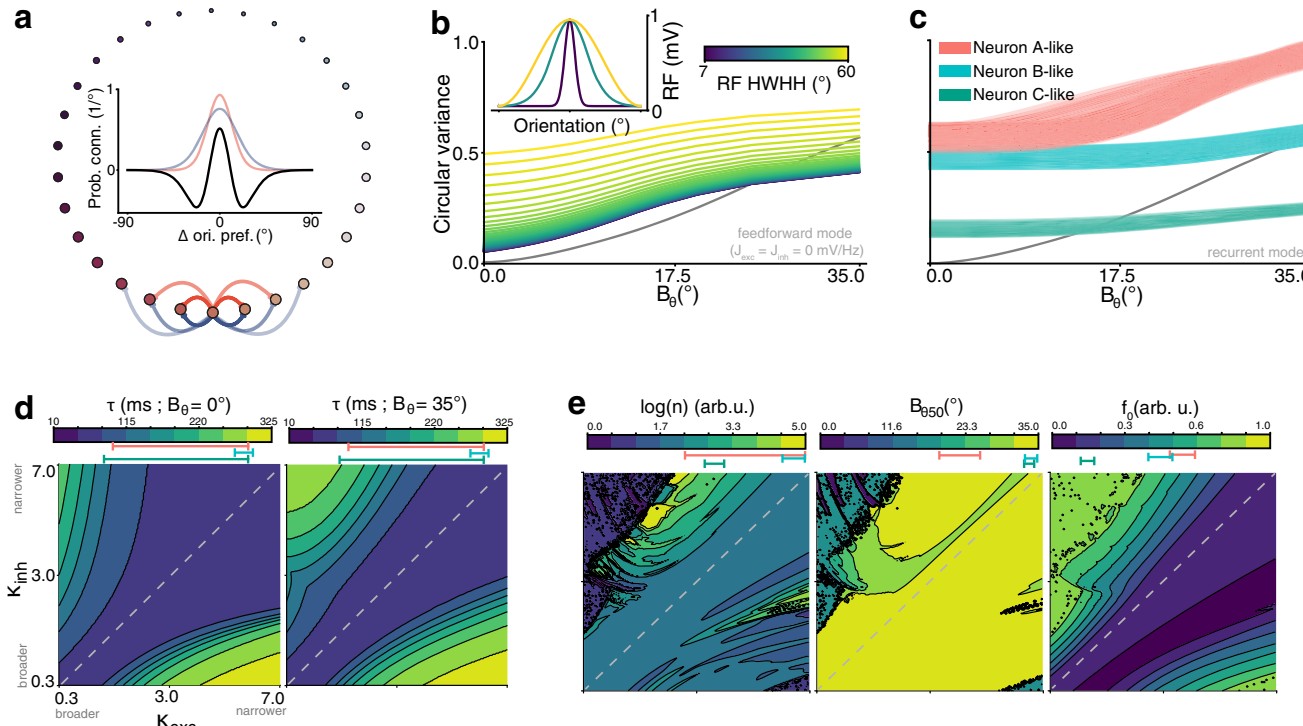

**Fig. 7 Recurrent interaction modeling can explain the existence of resilient and vulnerable neurons. a** Ring topology of the network, with the preferred orientation of each neuron. Inset: Recurrent connectivity profile for each neuron, computed as a difference (black) of excitatory (red) and inhibitory (blue) profiles, controlled by a measure of concentration $\kappa_{exc}$ and $\kappa_{inh}$, respectively. **b** VTF without recurrent connectivity (i.e., only inputs convolved with receptive fields) and with varying *RF*'s Half-Width at Half-Height (HWHH). The CV identity curve is shown in black. Inset shows examples of receptive fields *RF*. **c** VTF with recurrent connectivity, under two configurations retrieved by searching for VTF parameters close to those of neuron A, B (Fig. 2b) and C (Supplementary Fig. 1). **d** Delay to half maximum firing rate of the model ($\tau$) for each connectivity profile, shown as a contour plot of $\kappa_{exc}$ and $\kappa_{inh}$. [5%;95%] range of the parameters corresponding to the VTFs in (**c**) are displayed below the scale bars. **e** VTF parameters obtained from the model for each connectivity profile, shown as a contour plot of $\kappa_{exc}$ and $\kappa_{inh}$. [5%;95%] range of the parameters of the VTFs shown in (**c**) are displayed below the scale bars.

Computationally, most existing models support the idea that processing orientation variance can be achieved solely with local V1 computations[10]. For instance, Goris et al.[9] reported that heterogeneously tuned V1 populations help encode the orientation distributions found in natural images and that this functional diversity could be accounted for by a linear-nonlinear (L-NL) model. While this could explain the diversity of tuning in our data (Fig. 2), we found that such a model failed to account for some types of modulations of the VTFs (Fig. 7b). Therefore, we employed a model designed to replicate intracortical cat V1 data[38] and demonstrated that it reproduces various VTFs and dynamics observed in our recordings. The model used here pools activity from multiple orientation-tuned units into a single neuron, which we interpreted as a local recurrent model. While our results do not require contributions from extrastriate regions to explain the observed results, the possibility of recurrence involving neurons outside V1 cannot be entirely ruled out at this time[52].

Our study confirms the findings in the anesthetized macaque literature[9] by identifying single-neuron variance modulations that serve as the basis for decoding orientation variance at the population level in V1. This suggests that a common mechanism may underlie this neural mechanism in both felines and primates, which is a fundamental computational requirement for the proper encoding of natural images in V1[53]. Although gain/variance V1 functions have been previously reported[17], we demonstrate a similar input-output relationship in the form of VTFs, that has the added benefit of characterizing and extrapolating variance modulations across the full dynamical range of V1 populations. Further, we finely analyzed the temporal component of the response that is absent from the literature. We propose that all these response properties can be linked to cortical layers, binding the idea that supragranular neurons with sharp tuning and slow dynamics[28,29] support the co-encoding orientation and its variance.

This leads to an interesting tie to Bayesian inference, namely under the specific case of predictive coding[34], that canonically assigns (inverse) variance weighting of cortical activity to supragranular recurrent connectivity[6,8], without the need for extrastriate computations. This is an interesting perspective that opens up a general interpretation of our results into the broader context of processing variance/precision/uncertainty at different scales of investigations. Extending the present results to other cortical areas or other sensory modalities would be a simple process, given the generative stimulus framework used here[18], which could yield pivotal new insights into our understanding of predictive processes in the brain.

## Methods

**Visual stimulation**. Motion Clouds are generative model-based stimuli[18] that allow for fine parameterized control over naturalistic stimuli[54], which is a desirable trait when probing sensory systems under realistic conditions[21]. They are mathematically defined as band-pass filtered white noise stimuli, whose filters in Fourier space are defined as a parameterized distribution in a given perceptual axis (here, only orientation, but can be extended to speed[55] and scale[56]). Thus, the Motion Clouds presently used are fully characterized by their mean orientation and their

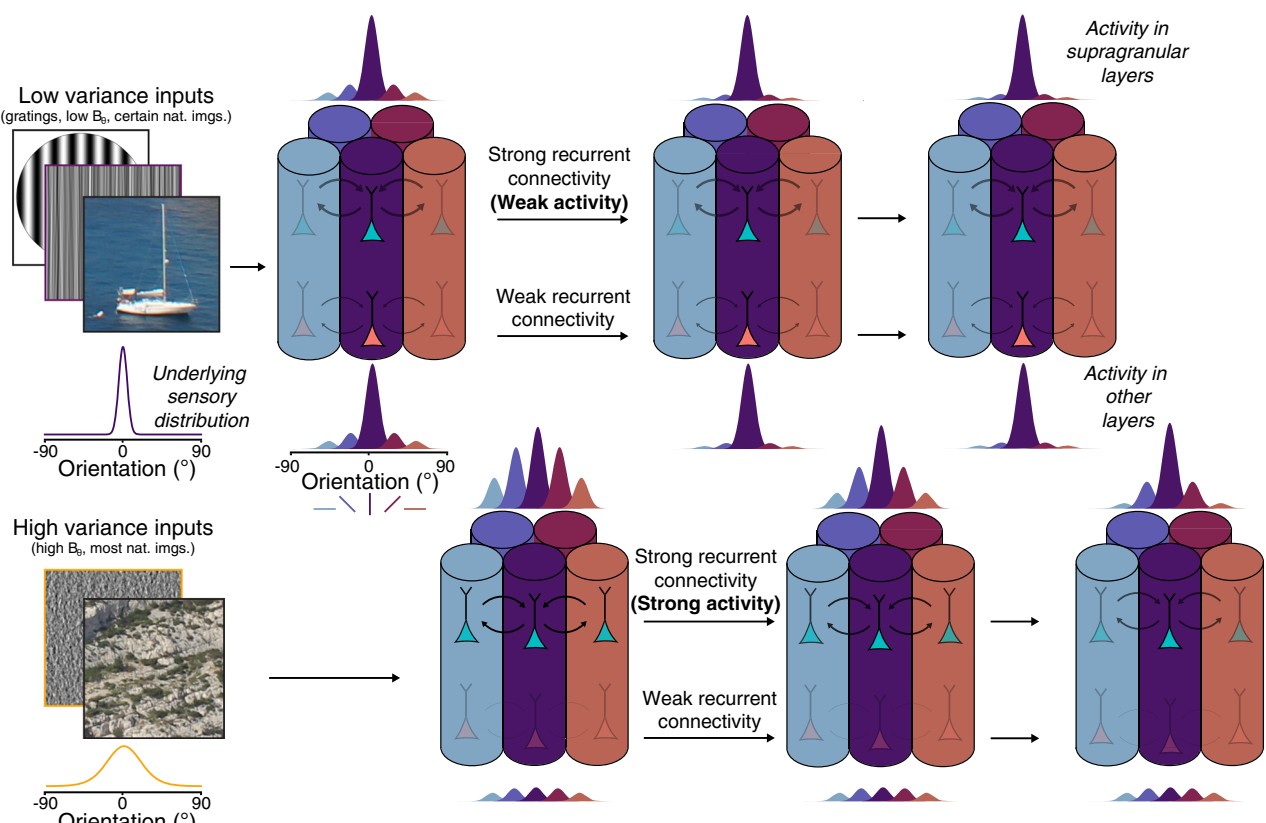

**Fig. 8 Summary of the findings.** The top row is a representation of a set of orientation-selective units (here, columns) processing low-variance inputs, while the bottom row schematizes the processing of high-variance inputs. In the case of low-variance inputs to V1, the underlying sensory distribution is sharp in the orientation space, driving mostly a single orientation-selective unit that processes orientation in a fast feedforward manner. The feature encoded by this activity then stays stable through time (from left to right). For inputs of higher orientation variance, the sensory input is broadly distributed in the orientation space, which drives many dissimilarly tuned units, thus recruiting slow recurrent interactions. The quality of feature encoding progressively increases through time, as recurrent interactions perform computations to represent the most salient oriented feature in the input.

orientation variance, such that a given stimulus $S$ can be defined as:

$$S = \mathcal{F}^{-1}(O(\theta, B_\theta)) \quad (1)$$

where $\mathcal{F}$ is the Fourier transform and $O$ the orientation envelope, characterized by its mean orientation $\theta$ and its orientation bandwidth $B_\theta$. For $B_\theta < 45.0°$, $B_\theta = 1/\sqrt{\kappa}$, where $\kappa$ is the concentration parameter of a von Mises distribution, and hence approximates the standard deviation[57]. It thus serves as a measure of the orientation variability in the pattern, and as such, we used the term variance to describe it throughout the text. A total of 96 different stimuli were generated, with 12 mean orientations $\theta$ ranging from 0 to $\pi$ in even steps, and eight orientation variance $B_\theta$ ranging from $\approx 0$ to $\pi/5$ in even steps. The orientation envelope is a von Mises distribution:

$$O(\theta, B_\theta) = \exp\left\{\frac{\cos(2(\theta_f - \theta))}{4 \cdot B_\theta^2}\right\} \quad (2)$$

where $\theta_f$ is the angle of the frequency components of the envelope in the Fourier plane, which controls the spatial frequency parameters of the stimuli, set here at 0.9 cycle per degree. The stimuli were drifting orthogonally in either direction with respect to the mean orientation $\theta$ at a speed of 10°/s, which is optimal to drive V1 neurons[58]. For the range of values of $B_\theta$ considered here, the orientation envelope approximates a Gaussian distribution and $B_\theta$ is thus a measure of the variance of the orientation content of the stimuli.

All stimuli were generated using open-source Python code (see Additional information) and displayed using Psychopy[59]. Monocular stimuli were projected with a ProPixx projector (VPixx Technologies Inc.) onto an isoluminant screen (Da-Lite©) covering 104° × 79° of visual angle. All stimuli were displayed for 300 ms, interleaved with a mean luminance screen (25 cd/m²) shown for 150 ms between each trial. Trials were fully randomized, and each stimulus (a unique combination of $\theta \times B_\theta \times$ drift direction) was presented 15 times. Stimuli were shown at 100% contrast, meaning that as $B_\theta$ increased, the amount of orientation energy at median orientation $\theta$ decreased, and conversely for off-median orientations (as illustrated in Fig. 1b). This differs from manipulating the contrast, which would reduce the orientation energy at all orientations.

**Surgery.** Experiments were conducted on three adult cats (3.6–6.0 kg, 2 males). All surgical and experimental procedures were carried out in compliance with the guidelines of the Canadian Council on Animal Care and were approved by the Ethics Committee of the University of Montreal (CDEA #20-006). Animals were initially sedated using acepromazine (Atravet®, 1 mg/kg) supplemented by atropine (0.1 mg/kg). Anesthesia was induced with 3.5% isoflurane in a 50:50 mixture of $O_2:N_2O$ (v/v). Following tracheotomy, animals underwent artificial ventilation as muscle relaxation was achieved and maintained with an intravenous injection of 2% gallamine triethiodide (10 mg/kg/h) diluted in a 1:1 (v/v) solution of 5% dextrose lactated Ringer solution. Through the experiment, the expired level of $CO_2$ was maintained between 35 and 40 mmHg by adjusting the tidal volume and respiratory rate. Heart rate was monitored and body temperature was maintained at 37 °C by means of a feedback-controlled heated blanket. Lidocaine hydrochlorine (2%) was applied locally at all incisions and pressure points and a craniotomy was performed over area 17 (V1, Horsley-Clarke coordinates 4-8P; 0.5–2 L). Dexamethasone (4 mg) was administered intramuscularly every 12 h to reduce cortical swelling. Eye lubricant was regularly applied to avoid corneal dehydration.

**Electrophysiological recordings.** During each recording session, pupils were dilated using atropine (Mydriacyl) while nictitating membranes were retracted using phenylephrine (Mydfrin). Rigid contact lenses of appropriate power were used to correct the eyes' refraction. Anesthesia was changed to 0.5–1% halothane to avoid anesthesia-induced modulation of visual responses[47]. Finally, small durectomies were performed before each electrode insertion and a 2% agar solution in saline was applied over the exposed cortical surface to stabilize recordings. Linear probes (≈1 MΩ, 1x32-6mm-100-177, Neuronexus) were lowered in the cortical tissue perpendicularly to the pia, and extracellular activity was acquired at 30KHz using an Open Ephys acquisition board[60]. Single units were isolated using Kilosort 2[61] and manually curated using Phy[62]. Clusters with low amplitude templates or ill-defined margins were excluded from further analysis. The additional exclusion was performed if a cluster was unstable (firing rate below 5 spikes.s⁻¹ for more than 30 s), or if the neuron was not deemed sufficiently orientation selective ($R^2 < 0.75$ when fitted with a von Mises distribution). Passing that exclusion step,

all remaining neurons responded to Motion Clouds. Laminar positions were determined by the depth of the recording site with respect to the pia, which was then cross-validated by the evoked Local Field Potential (LFP) using sink/source analysis[63,64].

**Single neuron analysis.** Orientation tuning curves were computed by selecting a 300 ms window maximizing spike-count variance[65]. The firing rate was averaged across drift directions and a von Mises distribution[57] was fitted to the data:

$$f(\theta_k) = R_0 + (R_{max} - R_0) \cdot \exp\left\{\kappa \cdot (\cos(2(\theta_k - \theta_{pref})) - 1)\right\} \quad (3)$$

where $\theta_k$ is the orientation of the stimuli, $R_{max}$ is the response at the preferred orientation $\theta_{pref}$, $R_0$ the response at the orientation orthogonal to $\theta_{pref}$ and $\kappa$ a measure of concentration. To control for direction selectivity when averaging tuning curves across drift direction, we computed a direction selectivity index:

$$D_s = \frac{R_{pref} - R_{null}}{R_{pref}} \quad (4)$$

where $R_{pref}$ is the firing rate at the preferred direction (baseline subtracted) and $R_{null}$ is the firing rate at the preferred direction plus $\pi$. The quality of each tuning curve was assessed by computing a global metric, the circular variance (CV) of the unfitted data, which varies from 0 for perfectly orientation-selective neurons to 1 for orientation-untuned neurons[29]. It is defined as:

$$CV = 1 - \left| \frac{\sum_k R(\theta_k) \cdot \exp\{2i\theta_k\}}{\sum_k R(\theta_k)} \right| \quad (5)$$

where $R(\theta_k)$ is the response of a neuron (baseline subtracted) to a stimulus of angle $\theta_k$. The changes of CV as a function of $B_\theta$ were fitted with a Naka-Rushton function[22]:

$$f(B_\theta) = f_0 + f_{max} \frac{B_\theta^n}{B_\theta^n + B_{\theta 50}^n} \quad (6)$$

where $f_0$ is the base value of the function, $f_0 + f_{max}$ its maximal value, $B_{\theta 50}$ the stimulus' variance at half $f_{max}$ and $n$ a strictly positive exponent of the function.

The significance of the tuning to orientation was measured by comparing the unfitted firing rate at the preferred and orthogonal orientations across trials, using a Wilcoxon signed-rank test correct for continuity, and the maximum value of $B_\theta$ which yielded a significant result was designed as $B_{\theta max}$ (i.e., the maximum variance at which a neuron is still tuned). Shifts of the preferred orientation were evaluated as the difference of $\theta_{pref}$ between trials where $B_\theta = 0°$ and $B_\theta = B_{\theta max}$. The significance of the variation of the peak amplitude of the tuning curve was measured by comparing the unfitted firing rate at the preferred orientation between trials where $B_\theta = 0°$ and $B_\theta = B_{\theta max}$.

**Population decoding.** The parameters used to generate Motion Clouds were decoded from the neural recordings using a multinomial logistic regression classifier[30]. For a given stimulus, the activity of all the recorded neurons was a vector $X(t) = \begin{bmatrix} X_1(t) & X_2(t) & \cdots & X_{249}(t) \end{bmatrix}$, where $X_i(t)$ is the spike count of neuron $i$ in a time window $[t; t + \Delta T]$. The onset of this window $t$ was slid from $-200$ to 400 ms (relative to the stimulation time) in steps of 10 ms while $\Delta T$ was kept constant at 100 ms. It should be noted that merging neural activity across electrodes or experiments is a common procedure[66,67], which we validated in our data by verifying that the electrode or experiment which yielded the data could not be decoded from the neural activity (Supplementary Fig. 7). Mathematically, the multinomial logistic regression is an extension of the binary logistic regression[30] trained here to classify the spike vector $X(t)$ between $K$ classes. The probability of any such vector belonging to a given class is:

$$P(y = k | X(t)) = \frac{\exp\{\langle \beta_k, X(t) \rangle\}}{\sum_{k'=1}^{K} \exp\{\langle \beta_{k'}, X(t) \rangle\}} \quad (7)$$

where $\langle \cdot, \cdot \rangle$ is the scalar product over the different neurons, $k = 1, \ldots, K$ is the class out of $K$ possible values and $\beta_k$ are the coefficients learned during the training procedure of the classifier. Several decoders were trained with classification tasks: decoding orientation $\theta$ ($K = 12$, Fig. 5), decoding orientation variance $B_\theta$ ($K = 8$, Supplementary Fig. 8) or both ($K = 12 \times 8 = 96$, Fig. 6). All meta-parameters were controlled, showing that the decoding performances stem mainly from experimental data rather than fine-tuning of the decoder parameterization (Supplementary Fig. 6). For all decoding experiments reported, we used integration window size $\Delta T = 100$ ms, penalty type $= \ell_2$, regularization strength $C = 1$. and train/test split size $= 0.15$.

The performance of all decoders was reported as the average accuracy across all classes $K$, known as the balanced accuracy score[68]. The accuracy for each specific class $k$ can also be reported in the form of a population tuning curve, in which the likelihood of decoding each possible class $K$ is given by equation (7). The significance of differences between two neuron groups was reported only when two consecutive time steps, i.e., 20 ms or more, exhibited significant differences. To estimate the time course of the decoders, they were fitted in the [0; 300] ms range

with a sigmoid function:

$$\sigma = max_{acc}\left(\frac{1}{1 + e^{-k\tau}}\right) + min_{acc} \quad (8)$$

where $max_{acc}$ and $min_{acc}$ are respectively the maximum and minimum accuracies of the decoder, $k$ the steepness and $\tau$ the time constant of the function. To perform decoding on the same number of vulnerable or resilient neurons, we randomly picked replacement groups of 100 neurons and bootstrapped this process five times.

As the neurons were clustered into two populations for comparison purposes (Fig. 4), we also reported the decoding accuracy based on a continuous vulnerability score (Fig. 6f). This score was computed as a sum of neuronal responses variables significantly different after the clustering, weighted by their mean Principal Component (PC-1 and PC-2) parameters:

$$\begin{aligned} score = &1 - W_1(B_{\theta 50}) + W_2(1 - \log(n)) + W_3(1 - f_0) + W_4(B_{\theta max}) \\ &+ W_5(1 - CV) + W_6(\text{early/late ratio}) + W_7(\text{delay}) \end{aligned} \quad (9)$$

where $W_i$ is a parameter yielded by the Principal Component Analysis corresponding to its associated neuronal response variable. Each variable is normalized, yielding a scalar score that varies between 0 (most resilient) to 1 (most vulnerable neuron). This score-based decoding was performed on groups of 100 neurons sorted by descending score and repeated a total of seven times on increasingly more vulnerable neurons (thus with an overlap of 20 neurons).

**Computational model.** We used a recurrent network of orientation-tuned neurons to model responses to increasing orientation variance $B_\theta$. The model presently used was first used to account for the intracortical activity in the cat primary visual cortex[36], although it was presently simplified as a center-surround filter in the orientation domain[29]. Notably, this network has been able to account for numerous experimental findings, including learning and adaptation of cortical neurons[37,38], whose implementations are similar to ours.

The model consisted of $N$ orientation-tuned neurons, evenly tiling the orientation space between $-\pi$ and $\pi$. Each neuron is modeled as a single passive unit whose membrane potential obeys the equation:

$$\tau \delta V / \delta T + V = V_{ff} + V_{exc} - V_{inh} \quad (10)$$

where $\tau$ is the membrane time constant and $V_{ff}, V_{exc}, V_{inh}$ are the synaptic potentials coming from the feedforward input, recurrent excitatory and recurrent inhibitory connectivity, respectively. The firing rate $R$ at time $t$ of each neuron is computed as an instantaneous quantity modulated by a gain $\alpha$:

$$R(t) = \alpha \cdot \max(V(t), 0) \quad (11)$$

For computational simplicity, the neurons had no spontaneous firing rate and $V$ was measured relative to the firing threshold. Each neuron could send mixed excitatory and inhibitory synaptic potentials to its neighbor, although this specific model has been reported to achieve similar behavior with separate units[38]. For each stimulus of main orientation $\theta$, the input to a cell with preferred orientation $\theta_{pref}$ is:

$$V_{ff}(\theta_{pref}) = J_{ff} \frac{e^{\kappa_{ff} \cdot \cos(2(\theta - \theta_{pref}))}}{2\pi I_0(\kappa_{ff})} \quad (12)$$

where $J_{ff}$ is the strength of the input and $I_0$ is the modified Bessel function of order 0. The right-hand side of the equation describes a von Mises with mean $\theta_{pref}$ and concentration $\kappa_{ff}$. This latter parameter is related to the orientation variance $B_\theta$, which was varied to yield a model's TVF $B_\theta / CV$ curves:

$$B_\theta = \sqrt{\frac{0.5 \arccos((\log(0.5) + \kappa_{ff}) / \kappa_{ff})}{2 \log(2)}} \quad (13)$$

a total of 20 $B_\theta$ spanning the same range used in the experiments were used, each with 32 different $\theta$ tiling a $[-75°; 75°]$ orientation space. The recurrent connectivity profile for excitatory ($C_{exc}$) and inhibitory ($C_{inh}$) synapses was controlled by separate von Mises distributions over the orientation space $\Theta$:

$$C_{exc}(\theta_{pref}) = \frac{e^{\kappa_{exc} \cdot \cos(2(\Theta - \theta_{pref}))}}{2\pi I_0(\kappa_{exc})} \quad (14)$$

$$C_{inh}(\theta_{pref}) = \frac{e^{\kappa_{inh} \cdot \cos(2(\theta - \Theta_{pref}))}}{2\pi I_0(\kappa_{inh})} \quad (15)$$

which are both used to describe an overall connectivity kernel:

$$C_{tot}(\theta_{pref}) = J_{exc} C_{exc} - J_{inh} C_{inh} \quad (16)$$

which followed a typical Ricker wavelet (or Mexican hat) shape (Fig. 7d). The overall activity of the network is then a weighted sum of the firing rates of all the neurons:

$$V_{exc} - V_{inh}(t) = \sum_{\Theta} C_{tot}(\theta_{pref}) \cdot R(t) \quad (17)$$

Parameterization of the model was done to match single V1 neuron recordings of anesthetized cats, in an experimental setup similar to the one used here[69].

The computational procedure to match experimental data was entirely done in a previous publication[38]. Briefly, it consisted in scanning a range of possible values for each parameter, then finding all possible combinations using a metric of likeliness to single grating response, time-to-peak, peak response and tuning width. The parameters yielded by this procedure were $\tau = 10.8$ ms; $\alpha = 10.6$ Hz/mV; $J_{ff} = 9.57$ mv/Hz; $J_{exc} = 1.71$ Hz/mV; $J_{inh} = 2.0178$ Hz/mV. For the feedforward mode of the model (Fig. 7b), $J_{exc}$ and $J_{inh}$ were set to 0 Hz/mV and the input was convolved with a receptive field:

$$RF = \frac{e^{\kappa_{RF} \cdot \cos(2(\theta - \Theta_{pref}))}}{2\pi I_0(\kappa_{RF})}) \qquad (18)$$

of which we reported the Half-Width at Half-Height, given by[70]:

$$HWHH = 0.5 \arccos(\frac{\log(0.5) + \kappa}{\kappa}) \qquad (19)$$

For the recurrent mode (Fig. 7c–e), the concentration measures of the recurrent connectivity profiles $\kappa_{exc}$ and $\kappa_{inh}$ were both varied from 0.35 to 7, in 200 even steps, and the input was not convolved with a receptive field.

**Statistics and reproducibility**. All data were analyzed using custom Python code. Statistical analysis was performed using non-parametric tests. Wilcoxon signed-rank test with discarding of zero-differences was used for paired samples and Mann–Whitney $U$-test with exact computation of the $U$ distribution was used for independent samples. Due to the impracticality of using error bars when plotting time series, colored contours are used to represent standard deviation values (unless specified otherwise), with a solid line representing mean values. For box-plots, the box extends from the lower to upper quartile values, with a solid white line at the median value. The upper and lower whiskers extend to respectively $Q1 - 1.5*IQR$ and $Q3 + 1.5*IQR$, where $Q1$ and $Q3$ are the lower and upper quartiles and $IQR$ is the interquartile range.

**Reporting summary**. Further information on research design is available in the Nature Portfolio Reporting Summary linked to this article.

## Data availability

Data used in the present study is publicly available in a Figshare repository[71]. Unprocessed electrophysiological recording files are available upon reasonable request to the corresponding author.

## Code availability

Custom Python code written for the present study is publicly available in a GitHub repository[72].

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

## Acknowledgements

The authors would like to thank Geneviève Cyr for her technical assistance, Bruno Oliveira Ferreira de Souza and Visou Ady for experimental advice, Louis Eparvier, Jean-Nicolas Jérémie and Salvatore Giancani for their comments on the manuscript, and Jonathan Vacher for fruitful exchanges on the formalization of the generation of synthetic images and for his contributions to related analysis of other neurophysiological recordings. This work was supported by the French government under the France 2030 investment plan, as part of the Initiative d'Excellence d'Aix-Marseille Université - A*MIDEX (AMX-21-RID-025), as well as by an ANR project "AgileNeuRobot" ANR-20-CE23-0021 to L.U.P, by a CIHR grant to C.C. (PJT-148959) and an École Doctorale 62 PhD grant to H.J.L.

## Author contributions

L.U.P., C.C., F.C., N.C., and H.J.L. designed the study. H.J.L., N.C. and L.I. collected the data. H.J.L. and L.U.P. analyzed the data. H.J.L. and L.U.P. wrote the original draft of the manuscript. All authors reviewed and edited the manuscript.

## Competing interests

The authors declare no competing interests.
