## [Peer Review File · Communications Biology]

Reviewers' comments:

Reviewer #1 (Remarks to the Author):

This is an interesting study looking at how the spread of orientations in a local patch of an image is encoded by neurons in primary visual cortex (in the anesthetized cat). This is an important topic that has been tackled before, for example by Goris and collaborators in two papers that are cited by the authors. The present study offers new data describing the tuning of neurons to mean and variance of an orientation distribution.

1) The data do not really show two classes of responses, but a continuum of behaviors. The decision to split the population into two groups is really arbitrary. Neither the analysis in Fig 4 or Fig S4 suggest that the data are best described by two classes of neurons.^[1]^[2] Also, why cluster only on the VTF responses? Why not include the entire responses (including the dynamics)?

2) One issue with the VTF analysis is that ignores how the strength of the responses change with increases in orientation variance. Yet, it is clear that a major contribution to the increase in circular variance could be due to lower responses at higher levels of orientation variance (eg, Fig 2A). It would be important to study more carefully how the responses are suppressed as orientation variance increases. Are the "resilient" cells those that have lower levels of suppression? — Incidentally, it is not clear if the circular variance of the responses is computed after subtracting the spontaneous rate of firing (Eqn 5 in the methods seems to suggest it is not).

3) Following up on the last point, how does the orientation bandwidth of the orientation tuning curve change as a function of orientation variance? It looks like neuron A in Fig 2A could still maintain a rather constant bandwidth despite the change in response magnitude.^[1]^[2] Suppose we have a cell with responses $R = A + B f(\theta)$, and that the only thing that changes with orientation variance is the magnitude of B. Would that cell be "resilient" or not (and why)?

4) How was stimulus contrast normalized as one increases orientation variance? The methods say that all stimuli were shown at 100% contrast. How is contrast defined for these class of stimuli? Did you normalize the resulting images between 0 and 1? Then the orientation energy at each orientation decreases as orientation variance increases.

5) It was not clear what the sizes of the training and testing sets were in the decoding analyses. The methods seem to indicate that you somehow optimized over these parameters as well. What is being reported in Fig 6? What were the final values used? ^[1]^[2]

6) The work of Goris and colleagues is mentioned briefly in the discussion, but it seems to me a more clear description of the similarities and differences in the results is needed here. Are the results largest confirmatory of what they found? If not, what is new in this study? Could you explain your data with the linear-nonlinear model of Goris et al as well? Why the need to invoke a new model?

Altogether, limitations in the analyses of the data, an (unjustified) description of the data as consisting of two classes of neurons, and a failure to explain how the work relates to prior studies, dampened the reviewer's enthusiasm for this study.

Reviewer #2 (Remarks to the Author):

In this manuscript, Ladret et al study neural responses in cat V1 to synthetic stimuli that vary in the concentration of image energy in the orientation domain. They build on previous work, but extend it in interesting directions. Specifically, while variants of the effects shown in Fig. 1 and 2 were both documented in Goris et al (2015), the effect of stimulus bandwidth on response dynamics (Fig. 3) is a new discovery. So is the proposal of two functional categories of cells ("vulnerable" and "resilient"), the differential distribution of these categories across cortical layers,

and the mechanistic account of this. I find these results valuable contributions to the literature. That said, I doubt the generality of the outcome of the decoding analyses (Fig. 5 and 6). I think these analyses should either be left out of the paper, or be improved.

1. The decoding analysis suggests that stimulus bandwidth cannot be decoded directly from the population response. How can that be? Increasing stimulus bandwidth increases the bandwidth of the population response profile and reduces its amplitude (this can be deduced from the example single unit responses in Fig. 2 and this pattern echoes the energy profile of the stimulus). A decoder that tries to estimate the dispersion of the population response should therefore in principle be able to estimate stimulus bandwidth. In practice, this may be difficult for small data-sets because neurons differ in their dynamic range. Due to the limited sample size, some stimulus orientations will elicit more population activity than others. This will complicate response dispersion estimation by a decoder. But if the population is large enough, these idiosyncrasies will disappear. The current finding seems to reflect a limitation of the specific decoder used in this analysis and perhaps of the idiosyncratic response distribution of this specific data set. I don't think it reflects a general property of V1 activity and I think it should not be presented as such.

2. To create populations of the same size, the authors opted to remove the least impactful vulnerable neurons from the population (Fig. 5c and related). I understand the logic behind this approach, but it creates an imbalance between both populations that is problematic. I believe it would be better to randomly sample, with replacement, a fixed number of neurons from both populations (e.g., 100), conduct the analysis on these populations, and then repeat for a different random sample. This is time-consuming, but it would yield a better estimate of the difference between resilient and vulnerable neurons than the current analysis.

3. Can the recurrent model reproduce the response latency differences for narrow and broad band stimuli shown in Fig. 3?

4. Minor point: there is a typo in the caption of Fig. 7e. k_{exc} should be k_{inh}

Dear Reviewers,

We would like to express our sincere gratitude for taking the time to review our manuscript. Your thorough reading and constructive criticism are invaluable to us, and we are pleased to reply to all the points you have raised in the following paragraphs. Updated figures can be found after the replies to your points (page 8-13).

Due to the need to rewrite the vast majority of our code to answer some of these questions, we have also taken the opportunity to improve the reproducibility of our results via better cross-validation and more stable model simulations. As such, we have made a few additional minor changes that are listed at the end of the document (page 14).

Reviewer #1:

This is an interesting study looking at how the spread of orientations in a local patch of an image is encoded by neurons in primary visual cortex (in the anesthetized cat). This is an important topic that has been tackled before, for example by Goris and collaborators in two papers that are cited by the authors. The present study offers new data describing the tuning of neurons to mean and variance of an orientation distribution.

1) The data do not really show two classes of responses, but a continuum of behaviors. The decision to split the population into two groups is really arbitrary. Neither the analysis in Fig 4 or Fig S4 suggest that the data are best described by two classes of neurons

We thank the reviewer for raising this important point on which we fully agree. The decision to split the population into two groups is indeed arbitrary and is done for argumentation's sake. Characterizing a continuum of behaviors, such as the one present in our data, is made easier when first characterizing the extremes of this continuum. This has often been done when describing novel visual responses, such as with simple/complex cells in V1 (Hubel and Wiesel 1962) or more recently with pattern/component cells in MT (Movshon 1985). Such method also also considerably simplifies the representation of high-dimensional decoding data, as is shown in Figures 5 and 6. This was indeed an implicit simplification in the manuscript, and we have now explicitly described our choice of focusing on two arbitrary classes of responses in lines 80-83.

Further, we now provide in Figure 6f (and lines 183-187) a continuum-based decoding approach. This consists of scoring each neuron based on the responses used in the K-means clustering (see Methods, lines 355-359), and reporting the maximum accuracy of a decoder with respect to the corresponding continuous vulnerable/resilient score. This new analysis supports in a clearer way our conclusion that neither class of neuron is better at encoding the mean of the variance of an orientation distribution, but that a higher resilience score correlates with a higher ability to co-encode orientation and variance simultaneously.

Also, why cluster only on the VTF responses? Why not include the entire responses (including the dynamics)?

Clustering is done on both the VTF and the dynamics, i.e., on all the parameters shown in Figure 4 (except Figure 4g and 4h, which are post-clustering comparisons). We agree that this requires clarification and have updated the related section (lines 73-74) accordingly. Furthermore, with the

continuous decoding scoring described in the previous question, the reader is now directed to a Methods section that describes in detail what response parameters are used in the scoring (lines 355-359), which should further clarify that the entire responses are used.

2) One issue with the VTF analysis is that ignores how the strength of the responses change with increases in orientation variance. Yet, it is clear that a major contribution to the increase in circular variance could be due to lower responses at higher levels of orientation variance (eg, Fig 2A). It would be important to study more carefully how the responses are suppressed as orientation variance increases. Are the “resilient” cells those that have lower levels of suppression? — Incidentally, it is not clear if the circular variance of the responses is computed after subtracting the spontaneous rate of firing (Eqn 5 in the methods seems to suggest it is not).

This is an excellent point. Our understanding here is that the reviewer first suggests that we compute a metric akin to a contrast-response function, which here would be a variance-response function that correlates input variance with the peak of the tuning curve’s firing rate. Such a function (with population histograms) is now shown in Supplementary Figure 2, which shows similar heterogeneity as with the VTFs. It is additionally compared between subpopulations of resilient and vulnerable neurons in the new Supplementary Figure 5, which shows no significant difference, except for the nonlinearity of the variance-firing rate function. However, this difference is implicitly present in the nonlinearity of the circular variance, as firing rate and circular variance are correlated due to the way this latter metric is computed (see Methods, lines 317-318; Eq 3).

Regarding the spontaneous rate, the circular variance of the responses is indeed computed after subtracting the spontaneous rate of firing. This is also the case for the tuning curves and the drift selectivity index (Eqn 3 and 4), and we have thus made explicit that all metrics are computed with baseline subtracted in lines 318, 321 and 324.

The reviewer makes a third very interesting point here by suggesting that there might be a correlation between resilience/vulnerability and suppression of firing rate. This would also make sense in interpreting the results as a recurrent population activity, in which resilient neurons would be either more or less suppressed depending on their I/E recurrent input. However, we did not observe any significant difference between the suppression of the baseline activity between vulnerable or resilient neurons.

3) Following up on the last point, how does the orientation bandwidth of the orientation tuning curve change as a function of orientation variance? It looks like neuron A in Fig 2A could still maintain a rather constant bandwidth despite the change in response magnitude. Suppose we have a cell with responses $R = A + B f(\theta)$, and that the only thing that changes with orientation variance is the magnitude of B. Would that cell be “resilient” or not (and why)?

This is an important point, and as suggested by the reviewer we now compute such a variance/orientation bandwidth curve. We however prefer to use the VTF, as orientation bandwidth is a metric computed using a fit, rather than directly on data like circular variance, thus constituting a second-order analysis which is susceptible to fitting artifacts (Mazurek et al, 2014, Front. Neur. Circ.). Performing numerous (249) fits here would amplify biases and distortions, leading to

potentially misleading conclusions. We believe utilizing raw data ensures more reliable and accurate insights, especially as we later relate circular variance to population tuning curve (starting in Figure 5b). Second, orientation bandwidth measures the strength of tuning around the peak of the orientation response, whereas circular variance provides a global metric and thus involves less prior hypothesis on selecting which part of the tuning curve should be included in the analysis (see Carandini et al, 2002, J. Neurosci., which provides a comparison and correlation between the two metrics).

We do agree that this an interesting analysis which should be present in the article. As with the variance/firing rate curves, we now provide variance/half-width at half height (HWHH) curves for Neuron A and B in Supplementary Figure 2, as well as related population histograms. CV and HWHH curves tell a similar story, and we believe that, as CV includes already a ratio of firing rate at preferred orientation, maximum firing rate curves do not provide more information to the reader. This is confirmed by the post-splitting difference in Supplementary Figure 5.

Finally, cells that would fit the model suggested by the reviewer would be those with a linear VTF function, which is indeed supported by a Naka-Rushton function description (Figure 2b shows how resilient neurons are characterized by a lower $\log(n)$ value). In the case of $R = A + B f(\theta)$, if only B changed, then the circular variance would be constant, and the cell considered resilient. However, as the Naka-Rushton function is more flexible for the others, non-linear, cases (Figure S2), we have preferred to keep it as our choice of response descriptors. We now extend the Bayesian Information Criterion plot of Figure S2 to show that this extends well to HWHH and maximum response variables.

4) How was stimulus contrast normalized as one increases orientation variance? The methods say that all stimuli were shown at 100% contrast. How is contrast defined for these class of stimuli? Did you normalize the resulting images between 0 and 1? Then the orientation energy at each orientation decreases as orientation variance increases.

This indeed requires clarification in the article, and we have rewritten the related section, lines 285-288. The reviewer is correct in their understanding, as contrast is defined much like in a grating, i.e., as the difference between the lowest and highest luminance of the stimuli (Michelson-type). As such, stimuli were normalized between minimal and maximum amplitude used by the software (here, -1 for dark and 1 for white in Psychopy). As illustrated in Figure 1b, this decreases the number of components at peak orientation as the orientation variance increases, whilst the opposite occurs for off-peak orientations. In terms of a Fourier transform, this means that MotionClouds contain less orientation energy/information as their variance ($B\theta$) increases, yet the global energy remains constant, which is the desired effect.

This notably differs from just decreasing contrast, which reduces the overall energy of the image, rather than changing its distribution between peak/off-peak orientations. However, both these sources of uncertainties seem to share a common processing mechanism according to Henaff et al, 2020.

5) *It was not clear what the sizes of the training and testing sets were in the decoding analyses. The methods seem to indicate that you somehow optimized over these parameters as well.*

We now explicitly report which parameters were used in the relevant Methods section (lines 347-348), as that indeed wasn't clear before. Regarding the reviewer's specific question, the testing set represents 15% of the total data, whilst the remaining 85% was used to train the decoders, which yielded the maximum decoding accuracy (Supplementary Figure S6d). Concerning the rest of the parameters, they were chosen after a parameter scan, which is reported in the Supplementary Figure S6, and likewise indicated as used for all decoders.

What is being reported in Fig 6? What were the final values used?

We are unsure about whether the reviewer would like a more detailed description of Figure 6, or if they would like to know which parameters were used.

In the former case, if the reviewer wishes for a more detailed explanation of Figure 6, we would refer him to our reply to reviewer #2's first point, in which we carefully detail our interpretation of the decoding results.

In the latter case, if the reviewer wishes to know which parameters were used, all decoders used the combination of parameters now reported in lines 347-348, i.e. 15% testing size, 100ms time window, L2 penalty and C=1 regularization. These parameters were chosen based on a range of values tested in Supplementary Figure 6, which shows that optimal parameters for decoding orientation also yield optimal performances for decoding orientation or orientation x variance.

6) *The work of Goris and colleagues is mentioned briefly in the discussion, but it seems to me a more clear description of the similarities and differences in the results is needed here. Are the results largest confirmatory of what they found? If not, what is new in this study?*

This is a fundamental point for which we thank the reviewer. The work of Goris and colleagues (i.e., Goris et al. 2015 and its reanalysis in Henaff et al. 2020) is essential to our own research, as these two articles make up the bulk of the literature on the electrophysiological basis of variance processing in V1. We have fully rewritten the paragraph that the reviewer refers to (lines 240-257) to properly compare Goris' research to ours, namely:

- Starting by disclaiming that our results are confirmatory of theirs, as we report a diversity of tuning of V1 neurons that plays a functional role in the encoding of a variance-changing sensory input (akin to Goris et al 2015). While their decoding approach and ours are methodologically different, both yield aligning messages. We argue that the similarity between both results is an interesting result in itself, that could serve to infer a shared mechanism for processing input variance across mammals with a columnar V1 organization.

- We then detail that our analysis goes into finer detail at the single-cell level, by

- a) providing a novel general function that relates input variance to neuron tuning variance (VTFs), which paves the way for many further studies, much like contrast-response functions.

- b) Having a dynamical component in the analysis, which is absent from Goris' work (as also pointed out by reviewer #2's introductory paragraph), but direly needed when studying the temporal evolution of population activity.

- We end by arguing that all of these results depend on layer-wise computations, and thus different connectivity, which is a major finding of our article.

Could you explain your data with the linear-nonlinear model of Goris et al as well? Why the need to invoke a new model?

The fact that we invoke a new model is very important to us, and we thank the reviewer for this question. This now constitutes the introductory point of our discussion with respect to Goris et al.'s articles (lines 240-248).

We did try to use a model that is computationally equivalent to that of Goris et al. in Figure 7b, showing that such a gain-control model could indeed only change the gain of the VTFs. We do not disagree with the fact that an L-NL model can indeed produce the diversity of tuning curves we also have in our results, but given its failure at explaining VTFs, we chose another approach.

A second reason for a recurrent model is, given that our "orientation x variance" decoding yields better results from the supragranular layers of V1 (backward inductive reasoning from Figure 6 to 4), we wished to build a model that would be based on the recurrence-heavy connectivity of these cortical layers. Such a recurrent model allows us to account both for the VTF and for dynamics (the latter are now presented in Figure 7d after reviewer #2's 3rd point). We, however, do not seek to completely separate our work from the literature and argue (lines 245-248) that locally recurrent models can be expressed in computationally similar feedforward/feedback models. For example, the V1 simple/complex cells behaviors can be explained both by a canonical Hubel and Wiesel feedforward model (1962) but also by recurrent supragranular amplification model (Chance et al., 1999).

We would like to thank you once again for your questions, which we hope to have answered in a satisfying manner. Your feedback has significantly raised the quality of the single-neurons results, whilst also referencing our article better with respect to the literature, which was sorely needed.

Reviewer #2:

In this manuscript, Ladret et al study neural responses in cat V1 to synthetic stimuli that vary in the concentration of image energy in the orientation domain. They build on previous work, but extend it in interesting directions. Specifically, while variants of the effects shown in Fig. 1 and 2 were both documented in Goris et al (2015), the effect of stimulus bandwidth on response dynamics (Fig. 3) is a new discovery. So is the proposal of two functional categories of cells (“vulnerable” and “resilient”), the differential distribution of these categories across cortical layers, and the mechanistic account of this. I find these results valuable contributions to the literature. That said, I doubt the generality of the outcome of the decoding analyses (Fig. 5 and 6). I think these analyses should either be left out of the paper, or be improved.

1. The decoding analysis suggests that stimulus bandwidth cannot be decoded directly from the population response. How can that be? Increasing stimulus bandwidth increases the bandwidth of the population response profile and reduces its amplitude (this can be deduced from the example single unit responses in Fig. 2 and this pattern echoes the energy profile of the stimulus). A decoder that tries to estimate the dispersion of the population response should therefore in principle be able to estimate stimulus bandwidth. In practice, this may be difficult for small data-sets because neurons differ in their dynamic range. Due to the limited sample size, some stimulus orientations will elicit more population activity than others. This will complicate response dispersion estimation by a decoder. But if the population is large enough, these idiosyncrasies will disappear. The current finding seems to reflect a limitation of the specific decoder used in this analysis and perhaps of the idiosyncratic response distribution of this specific data set. I don't think it reflects a general property of V1 activity and I think it should not be presented as such.

We thank the reviewer for this important point, on which we fully agree. We would also expect that the bandwidth of the population could be read out from a sufficiently large number of neurons, and it is indeed likely that the small size of the sample limits our conclusion (now mentioned lines 148-153).

Biologically, one could counter-argue that this limitation might hold even at high population count, as the low firing rate at preferred orientation for high stimulus bandwidth and low firing rate at unpreferred orientation for low stimulus bandwidth renders the scheme of encoding bandwidth through firing rate difficult. As such, it would make little intuitive sense for the cortex to have specialized units for the encoding of variance independently of any sensory feature, given the ambiguous nature of the message encoded by the firing rate. This would require « signing » the neuronal message passing through orientation-specific connectivity, which is supported only at the short-range distance in V1 (see Chavane et al., 2011). Other spiking encoding schemes, such as encoding input variance through firing rate variance, are now properly mentioned in the discussion (lines 249-257), which serves to frame the currently limited decoding approach in a global context with plausible alternatives.

To reflect this general point, we are now presenting decoding results with more nuance in the manuscript. Rather than affirming the encoding schemes under study are indeed present in V1, we instead only report that such decoding is possible. For instance, “resilient neurons can co-encode both orientation and its variance simultaneously” would now be written as “orientation and its

variance can be co-decoded simultaneously from resilient neurons” (see lines 139-141; 152-153; 177-182, as well as the legend of Figures 5 and 6).

Overall, we would be very much in favor of keeping this section of the text because it provides a functional rationale for the presence of multiple single neuron behaviors. One future direction of our research that is currently underway includes high-density recordings, and we would very much like to establish this decoding principle in the literature to see whether it will hold for higher neuron counts.

2. To create populations of the same size, the authors opted to remove the least impactful vulnerable neurons from the population (Fig. 5c and related). I understand the logic behind this approach, but it creates an imbalance between both populations that is problematic. I believe it would be better to randomly sample, with replacement, a fixed number of neurons from both populations (e.g., 100), conduct the analysis on these populations, and then repeat for a different random sample. This is time-consuming, but it would yield a better estimate of the difference between resilient and vulnerable neurons than the current analysis.

This is indeed an important limitation on the comparison between resilient and vulnerable neurons. We have rewritten the entirety of the analysis pipeline to implement the reviewer’s suggestion of bootstrapping with replacement across pools of neurons. This is now mentioned lines 132-135, in the text of Figure 5 and in the Methods section, lines 354-355.

Briefly, a random selection with replacement in both populations (n=100, repeated 5 times) was used before decoding. We thus no longer report the 5-fold cross validation from a single set of neurons (split 5 different ways), but rather the mean of this 5-fold validation for each 5 resampling. This does not change our general interpretation of the results but does yield a better estimate of the difference between the two populations, as the reviewer suggested. All the figures related to the decoding section have been updated accordingly.

One novel result that was previously absent is a significant difference between resilient and vulnerable neurons when decoding orientation (Figure 5c). This difference shows up on three time windows and does not cover the entirety of the time course, and is much smaller than the difference observed in the orientation x variance co-encoding. This is now discussed lines 132-141, where such result is weighted by the fact that decoding tuning curves are similar for the two populations.

3. Can the recurrent model reproduce the response latency differences for narrow and broad band stimuli shown in Fig. 3?

As the reviewer rightfully pointed out in his introductory paragraph, the response latency is one key novelty compared to Goris et al.’s research., and one that the model needs to be able to reproduce. We had elected not to present it out of clarity’s sake in the first draft of the article, but now report that the model does produce these responses latencies in Figure 7d (and associated text lines 213-215). Rather than showing a histogram as in Figure 3, we elected to keep the same representation as Figure 7d, which has the added benefit of linking VTFs and responses latencies with the same E/I recurrence parameters. As the model is based on Leaky Integrate-and-Fire

neurons, we reported the time constant of the population, rather than the time to maximum firing rate, which yields a more precise approximation of the dynamics of the population.

To fit this new data into the Figure, we have elected to remove the former Figure 7a, which showed the type of inputs that was given to the network. We believe this was not necessary to understand how the model worked, but have nonetheless updated lines 197-199 as a text alternative for this former subfigure.

4. Minor point: there is a typo in the caption of Fig. 7e. k_{exc} should be k_{inh}

We thank the reviewer for their keen eye, and have fixed this typo whilst also sweeping throughout the article to ensure no other similar issues were present in the text nor figures.

We would like to thank you once again for your questions, which we hope to have answered in a satisfying manner. Your feedback has significantly raised the quality of the decoding analysis, and in the process has allowed us to write a better and clear code for all readers to access. Asking for dynamical analysis of the model also allows us to finish our writing with a coherent conclusion, which improves greatly the quality of the article.

List of updated figures

Updated Figure 1:

- Added y-axis label and ticks to Figure 1 b's histograms.

Updated Figure 4:

- Swapped the order of vulnerable/resilient groups to be consistent throughout the figure, with vulnerable neurons shown on the left and resilient neurons on the right of each subfigure.

Updated Figure 5:

- Updated decoding results to reflect the new bootstrap with resampling selection method. This notably adds a significant difference in Figure 5c, which we discuss in the reply to reviewer 2 question 2 and in the text (lines 136-141)

Updated Figure 6:

- As for Figure 5, updated this figure to reflect the new bootstrap with resampling method
- This does not change the interpretation of the results
- Figure 6e now reports a θ ; B_θ map, rather than a $\Delta\theta$; B_θ map. This means that we no longer report the orientation encoding error as a function of the variance, but rather the orientation decoded as a function of the variance decoded, which is much simpler to visualize and understand.
- Added Figure 6f, which shows the continuous decoding results. This is detailed in the reply to reviewer 2 question 1.

Updated Figure 7:

- Removed former Figure 7a in favor of former Figure 7b (this now-absent subfigure is instead discussed in the text, lines 197-199)
- Added the dynamical results of the method in Figure 7d, which we discuss in reply to reviewer 2 question 3

Updated Figure 8:

- Made the layout of the figure more compact

Updated Sup. Figure 2:

- Added the variance-HWHH and variance-maximum firing rate functions, histograms and goodness-of-fit based on the reply to reviewer 1 questions 2 & 3.

New Sup. Figure (now Sup. Figure 5 in the manuscript):

- Added this supplementary figure based on the reply to reviewer 1 questions 2 & 3, which describes the post-clustering separation of the new variance-HWHH and variance-maximum firing rate functions.

Minor points that were not asked by the reviewers:

- Formatting now follows the journal guidelines.
- Y-axis ticks and legend have been added to Figure 1b.
- We have swapped the order of resilient and vulnerable data in Figure 4 to be consistent throughout the figure, by showing vulnerable on the left and vulnerable on the right in each subplot (this was done the opposite way only in 4d and 4g).
- We have updated the model's code for better numerical stability and Python 3.10.0 compatibility, which, over the 400^2 simulations, ends up yielding slightly different range of parameters for the three example VTFs shown in Figure 7d. Thus, the scale bars above Figure 7e have had their extent updated to reflect these new results. This does not change the interpretation of the results, as all three types of responses still have the same distinguishable range of parameters, except for $\log(n)$, which already had overlapping domains in the previous implementation of the model.
- We have changed the layout of Figure 8 to be more compact and more readable. The text "Strong recurrent connectivity (weakly activated)" has been changed to "Strong recurrent connectivity (weak activity)" (and conversely for strongly activated) in order to fit this more compact layout.
- We have made an effort to improve the conciseness of the text throughout the article, which improves its readability without changing its content. As these are minor but ubiquitous phrasing changes, we have not highlighted them in the revised version of the manuscript.

REVIEWERS' COMMENTS:

Reviewer #2 (Remarks to the Author):

The authors have addressed my concerns.

I have also carefully re-read the rebuttal letter and revised manuscript and I find that the authors have done a satisfactory job to address reviewer 1's concerns.